# Coordinated Search for Symbolic Formulas of Complex Network Dynamics

## Abstract

Distilling the dynamics of complex networks into symbolic formulas is a fundamental goal in science. However, existing neural symbolic regression methods often search for node (self-evolution) and edge (interaction) dynamics independently. This can lead to overfitting, where errors in one component are compensated for by an overly complex expression for the other, yielding uninterpretable and non-generalizable models. We introduce **Coordinated Genetic Search (CGS)**, a novel algorithm that discovers these symbolic expressions cooperatively. CGS first trains a disentangled neural proxy model to provide reliable references and denoised, interpolated trajectories. It then co-evolves two populations of symbolic expressions—one for node and one for edge dynamics—by strategically prioritizing the evolution of the population that deviates most from its neural reference. This coordinated process prevents overfitting and steers the search toward a balanced, accurate solution. Evaluated on synthetic dynamics and a real-world disease spreading dataset, CGS significantly surpasses previous approaches in formula recovery and prediction accuracy, consistently discovering simpler, more generalizable, and more physically faithful symbolic models.

## 1 Introduction

Complex networks (Gerstner et al., 2014; Gao et al., 2016; Bashan et al., 2016; Newman et al., 2011) are the fabric of our interconnected world, from the intricate web of social interactions (Kitsak et al., 2010) and the pathways of global pandemics (Pastor-Satorras & Vespignani, 2001) to the complex wiring of the human brain (Laurence et al., 2019; Wilson & Cowan, 1972). Understanding how these networks evolve is a central challenge in modern science (Zang & Wang, 2020; Murphy et al., 2021; Gao & Yan, 2022). The ultimate goal is not just to observe their dynamics, but to distill them into concise, sym-

Table 1: CGS produces simpler expressions with better generalization for SIS dynamics.

| **Overfitted Expression** |
|---|
| **SymDL (Search Separately):** $\dot{\mathbf{x}}_v(t) = -0.6017\mathbf{x}_v(t)\sin(\cos(\mathbf{x}_v(t) - 0.5178)) + \sum_{u \in N_v}[(0.9966 - 1.1798\mathbf{x}_v(t) + 0.1984\sin(\mathbf{x}_v(t)))\mathbf{x}_u(t)]$ |
| **Our Expression (CGS: Search Together)** |
| $\dot{\mathbf{x}}_v(t) = -0.48540\mathbf{x}_v(t) + \sum_{u \in N_v}(1 - \mathbf{x}_v(t))\mathbf{x}_u(t)$ |

bolic formulas (Pastor-Satorras et al., 2015; MacArthur, 1970; Kuramoto & Kuramoto, 1984; Gaucel et al., 2014; Kronberger et al., 2020; Brunton et al., 2016; Qian et al., 2022; d'Ascoli et al., 2024), i.e., the fundamental laws that govern their behavior. Like elegant physical equations, these symbolic expressions offer a clear window into the underlying mechanisms of a system, enabling us to predict its future and understand its core principles (Schmidt & Lipson, 2009; Petersen et al., 2019; Cranmer et al., 2020; Shi et al., 2023).

A promising approach is neural symbolic regression (Cranmer et al., 2020; Shi et al., 2023; Qian et al., 2022), which leverages deep learning models with inductive biases to discover symbolic laws. These methods are robust to noise and irregular sampling, making them more suitable for real-world data than traditional genetic programming (Gaucel et al., 2014; Kronberger et al., 2020) or sparse regression techniques (Brunton et al., 2016; Gao & Yan, 2022). However, a critical challenge remains. By incorporating the inductive bias, the neural network dynamics of a node in a complex network are a composite of two distinct processes: neural node dynamics (how a node acts on its own) and neural edge dynamics (how it interacts with its neighbors) (Liu et al., 2023). Existing methods often search

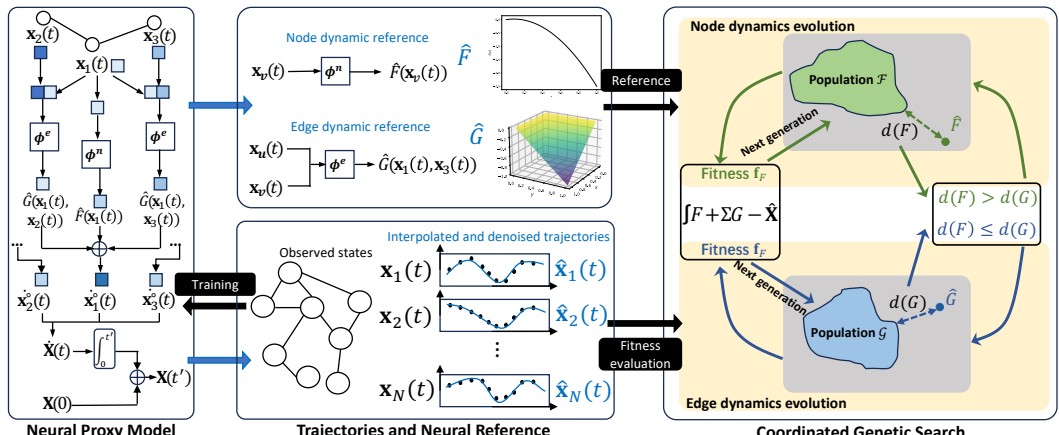

Figure 1: Overview of Coordinated Genetic Search (CGS). A neural proxy model provides references $(\hat{F}, \hat{G})$ and trajectories $(\hat{X})$ to guide the co-evolution of symbolic populations for node and edge dynamics. The search prioritizes the population deviating most from its reference to find accurate symbolic expressions $(F^*, G^*)$.

for these components' formulas independently (Cranmer et al., 2020; Shi et al., 2023), which can lead to a critical form of overfitting: errors in one component are compensated for by an inaccurate expression for the other. This results in models that are neither interpretable nor generalizable, as illustrated in Table 1.

To address this, we introduce **Coordinated Genetic Search (CGS)**, a novel search algorithm that co-evolves symbolic expressions for both node and edge dynamics. CGS operates on the principle that these components must be discovered cooperatively to avoid overfitting, where errors in one component are compensated for by an overly complex expression for the other. The algorithm first trains a disentangled neural proxy model to generate reliable references for each neural dynamics component and to provide denoised, interpolated trajectories for fitness evaluation. It then maintains two distinct populations of symbolic expressions—one for node dynamics and one for edge dynamics—and coordinates their evolution by strategically prioritizing the population that deviates most from its neural reference. This coordinated process, guided by neural references and evaluated against the interpolated trajectories, steers the search toward simpler, more generalizable, and more faithful symbolic models. We evaluate CGS on various synthetic dynamics and a real-world disease spreading dataset, demonstrating that it significantly surpasses previous approaches in both formula recovery and prediction accuracy.

**Notations**   Matrix, vector, and scalar are denoted as bold capital letters $\mathbf{X}$, bold lowercase letters $\mathbf{x}$, and lowercase letters $x$, respectively. The element in $i$-th row and $j$-th column of matrix $\mathbf{X}$ is denoted as $\mathbf{X}_{ij}$. The $v$-th row of matrix $\mathbf{X}$ is denoted as $\mathbf{x}_v$. A complex network is denoted as $\mathbf{G} = (\mathsf{G}, \mathbf{X}(t), t \in \mathcal{T})$. $\mathsf{G} = (V, E)$ denotes a network with node set $V$ and edge set $E$. $\mathbf{X}(t) = [\mathbf{x}_1(t)^\top, \cdots, \mathbf{x}_N(t)^\top]^\top \in \mathbb{R}^{N \times d}$ is a $d$-dimensional node states of $N$ nodes at the timestamp $t$, and $\mathcal{T} = \{t_0, t_1, \cdots, t_{K-1}\}$ is the set of $K$ timestamps of complex network observations. $\dot{\mathbf{x}}(t)$ represents the time derivatives of $\mathbf{x}(t)$.

## 2   COORDINATED GENETIC SEARCH FOR SYMBOLIC REGRESSION

This section details our Coordinated Genetic Search (CGS) algorithm. We begin in Section 2.1 by introducing a neural proxy model that provides reliable references for node and edge dynamics, along with denoised and interpolated trajectories. Section 2.2 then explains how these neural references are used to coordinate the search, and Section 2.3 describes the fitness evaluation and evolution process. Finally, we compare CGS with existing methods in Section 2.4.

## 2.1 Neural Proxy Model

**Inductive Bias of Complex Network Dynamics**   The complex network dynamics is defined by the following differential ordinary equation:

$$\dot{\mathbf{x}}_v(t) = F\left(\mathbf{x}_v(t)\right) + \sum_{u \in N_v} a_{vu} G\left(\mathbf{x}_v(t), \mathbf{x}_u(t)\right), \tag{1}$$

In (1), $\dot{\mathbf{x}}_v(t)$ denotes the time derivative of $\mathbf{x}_v(t)$, $F(\mathbf{x}_v(t))$ denotes the node dynamics term of node $v$, which includes processes like influx, degradation, or reproduction. $G(\mathbf{x}_v(t), \mathbf{x}_u(t))$ is the edge dynamics describing the interactions between node $v$ and node $u$, accounting for processes such as spreading and competition. $G$ is shared across all edges in the network because of the universality in network dynamics (Barzel & Barabási, 2013; Gao et al., 2016). $a_{vu}$ is the weight of the edge between node $v$ and node $u$, and $N_v$ is the set of neighbors of node $v$. Given observations of the node states $\{\mathbf{X}(t)|t \in \mathcal{T}\}$, the symbolic regression of complex network dynamics (Gao & Yan, 2022) aims to find the symbolic expressions of function $F$ and $G$ in (1).

**Neural Proxy Model with Inductive Bias**   We follow the inductive bias from (1) to design a neural proxy model and train the model based on the observed trajectory. In the designed neural proxy, a graph neural network calculates the time derivative of the node state $\dot{\mathbf{x}}_v^\circ(t)$. An ODESolver then integrates this derivative to generate the full trajectory. Based on DNND (Liu et al., 2023), the time derivative $\dot{\mathbf{x}}_v^\circ(t)$ for node $v$ is designed as an encoder-decoder free architecture:

$$\dot{\mathbf{x}}_v^\circ(t) = \hat{F}(\mathbf{x}_v(t)) + \sum_{u \in N_v} \hat{G}\left(\mathbf{x}_v(t), \mathbf{x}_u(t)\right) \tag{2}$$

$$\text{where } \hat{F}(\mathbf{x}_v(t)){=}\phi^n(\mathbf{x}_v(t), t), \hat{G}\left(\mathbf{x}_v(t), \mathbf{x}_u(t)\right){=}\phi^e(\mathbf{x}_v(t), \mathbf{x}_u(t), t). \tag{3}$$

where $\phi^n$ and $\phi^e$ are two MLPs aligning with the node dynamics and edge dynamics in (1), respectively. In (2), the neural node dynamics $\phi^n$ captures the evolution of nodes influenced by their properties, and the neural edge dynamics $\phi^e$ captures the interactions between two end nodes of an edge. Therefore, proxy model of node $v$ is written as

$$f_\theta(\mathsf{G}, \mathbf{X}(t_0), t)_v = \text{ODESolver}(\dot{\mathbf{x}}_v^\circ(t), \mathbf{X}(t_0), t_0, t). \tag{4}$$

The alignment between neural dynamics (4) and dynamics formulation (1) enables better learning of complex network dynamics. To train the neural dynamics, we minimize the error between $f_\theta(\mathsf{G}, \mathbf{X}(t_0), t)_v, \forall v \in V$ and the observed trajectories $\{\mathbf{X}(t)|t \in \mathcal{T}\}$, i.e., $\min_\theta \sum_{v \in V, t \in \mathcal{T}} \|f_\theta(\mathsf{G}, \mathbf{X}(t_0), t)_v - x_v(t)\|_1$, with standard deep learning optimization techniques. After the training, we will use the estimated node dynamics and edge dynamics as references for the coordinated search and use the interpolated trajectory $\hat{\mathbf{X}}(t)$ from $f_\theta(\mathsf{G}, \mathbf{X}(t_0), t)_v, \forall v \in V$ as the signal for fitness evaluation.

## 2.2 Coordination via Neural References

Although the neural proxy model provides disentangled estimates for node dynamics ($\hat{F}$) and edge dynamics ($\hat{G}$), these components are not perfectly accurate. During training, errors in one component can be compensated for by the other, resulting in a model that fits the overall trajectory but misrepresents the individual dynamics. Therefore, using these neural components for direct supervision in separate searches for symbolic $F$ and $G$ would risk replicating this overfitting. Instead, CGS uses them as references to coordinate the evolution of two symbolic populations, ensuring a balanced search.

CGS maintains two populations of symbolic expressions: $\mathcal{F}$ for node dynamics and $\mathcal{G}$ for edge dynamics. To prevent one population from overfitting to compensate for inaccuracies in the other, their evolution is coordinated. At each step, CGS measures the deviation of each population from its respective neural reference:

$$d(\mathcal{F}) = \sum_{F \in \mathcal{F}} |F - \hat{F}|^2, \quad d(\mathcal{G}) = \sum_{G \in \mathcal{G}} |G - \hat{G}|^2, \tag{5}$$

where $|\cdot|$ is the average absolute error between two functions on randomly sampled points. The algorithm then prioritizes the evolution of the population with the larger distance to its reference. For instance, if $d(\mathcal{F}) > d(\mathcal{G})$, only the node dynamics population $\mathcal{F}$ is evolved. This strategy ensures a balanced search, preventing overfitting and improving the quality of the final symbolic expressions.

## 2.3 Fitness Evaluation and Evolution

While neural dynamics provide references for coordination, they are not precise enough for direct fitness calculation. Instead, CGS uses the denoised and interpolated trajectories from the proxy model as the ground truth for fitness evaluation.

To evolve the selected population, we assess the fitness of each candidate expression. The fitness of a symbolic node dynamics $F \in \mathcal{F}$ or a symbolic edge dynamics $G \in \mathcal{G}$ is calculated by pairing it with expressions from the other population and measuring the error against the interpolated trajectory:

$$\mathsf{f}_F = \text{Mean} \circ \text{BigK}\Big\{ \sum\nolimits_{v \in V, t \in T} -e\Big(f_{F,G}^{v,t}, f_\theta^{v,t}\Big)\Big| G \in \mathcal{G}\Big\}, \tag{6}$$

$$\mathsf{f}_G = \text{Mean} \circ \text{BigK}\Big\{ \sum\nolimits_{v \in V, t \in T} -e\Big(f_{F,G}^{v,t}, f_\theta^{v,t}\Big)\Big| F \in \mathcal{F}\Big\}, \tag{7}$$

$$\text{where} \quad f_{F,G}^{v,t} = \int_0^t \Big(F(\mathbf{x}_v) + \sum\nolimits_{u \in N_v} G(\mathbf{x}_v, \mathbf{x}_u)\Big)\, dt, \quad f_\theta^{v,t} = f_\theta\big(\mathsf{G}, \mathbf{X}(t_0), t\big)_v \tag{8}$$

Here, $e(\cdot, \cdot)$ is the error function, $T$ is the set of interpolated timestamps, BigK selects the $K$ best-performing pairs, and Mean averages their errors. A higher fitness value (lower error) indicates a better symbolic expression. The expressions in the selected population then undergo selection, crossover, and mutation to generate the next generation.

The complete CGS algorithm is detailed in Algorithm 1. The process begins by initializing populations $\mathcal{F}^{(0)}$ and $\mathcal{G}^{(0)}$. In each iteration, the algorithm decides which population to evolve based on (5). It then calculates fitness, checks for convergence, and applies genetic operators. The search terminates when a satisfactory solution is found or the maximum number of iterations is reached, returning the best-fit symbolic expressions $F^*$ and $G^*$.

---

**Algorithm 1** Coordinated Genetic Search for SR

---

**Require:** Neural dynamics $f_\theta$, node dynamics reference $\hat{F}$, edge dynamics reference $\hat{G}$, $K$ for calculating fitness, maximum iteration $M$, threshold $\epsilon$.
1: Initialize the node dynamics population $\mathcal{F}^{(0)}$ and edge dynamics population $\mathcal{G}^{(0)}$ with random symbolic expressions;
2: **for** $i = 1, 2, \cdots, M$ **do**
3:     Compute $d(\mathcal{F}^{(i-1)})$ and $d(\mathcal{G}^{(i-1)})$ using (5);
4:     **if** $d(\mathcal{F}^{(i-1)}) > d(\mathcal{G}^{(i-1)})$ **then**
5:         Calculate the fitness $\mathsf{f}_F$ of each expression $F$ in $\mathcal{F}^{(i-1)}$ using (6);
6:         **if** $\exists F \in \mathcal{F}^{(i-1)}, \mathsf{f}_F \leq \epsilon$, **break**;
7:         Select, cross, and mutate the expressions in $\mathcal{F}^{(i-1)}$ to generate the next population $\mathcal{F}^{(i)}$;
8:         $\mathcal{G}^{(i)} = \mathcal{G}^{(i-1)}$;
9:     **else**
10:        Calculate the fitness $\mathsf{f}_G$ of each expression $G$ in $\mathcal{G}^{(i-1)}$ using (7);
11:        **if** $\exists G \in \mathcal{G}^{(i-1)}, \mathsf{f}_G \leq \epsilon$, **break**;
12:        Select, cross, and mutate the expressions in $\mathcal{G}^{(i-1)}$ to generate the next population $\mathcal{G}^{(i)}$;
13:        $\mathcal{F}^{(i)} = \mathcal{F}^{(i-1)}$;
14:     **end if**
15: **end for**
16: $F^* = \arg\min_{F \in \mathcal{F}^{(i)}} \mathsf{f}_F$;
17: $G^* = \arg\min_{G \in \mathcal{G}^{(i)}} \mathsf{f}_G$;
18: **return** $F^*, G^*$.

---

## 2.4 Comparison

We compare SymDL (Cranmer et al., 2020), NASSymDL (Shi et al., 2023), D-CODE (Qian et al., 2022), TP-SINDy (Gao & Yan, 2022) and CGS in Table 3. These methods cater to different problem settings, utilizing distinct forms of input and output. SymDL and NASSymDL perform general symbolic regression, finding a function $y = f(x)$ from input-output pairs $(x_i, y_i)_{i=1}^{N}$. D-CODE focuses on *symbolic regression of dynamics*, taking a single trajectory $\{x(t)|t \in \mathcal{T}\}$ to output the governing ODE $\dot{x} = dx/dt$. TP-SINDy and CGS target *symbolic regression of complex network dynamics*, using multiple trajectories $\{\mathbf{X}(t)|t \in \mathcal{T}\}$ to output symbolic network dynamics $F$ and $G$.

We compare these algorithms in two aspects: proxy models and formula regression. Proxy models are trained to fit data and serve as a basis for deriving symbolic expressions. Formula regression directly extracts symbolic expressions from raw data or proxy models. For proxy model, SymDL uses graph networks (GN) with inductive bias, and NASSymDL employs neural architecture search (NAS) for skeleton search. D-CODE can incorporate any suitable regressor. CGS fits multiple trajectories using proxy, a graph neural ODE aligned with network dynamics for better generalization. SymDL and NASSymDL rely on potentially noisy estimated derivatives for dynamics fitting, whereas D-CODE and CGS train directly on raw observations for improved accuracy. TP-SINDy needs no proxy model.

In formula regression, methods using genetic search employ elementary operations (e.g., $+, -, \times, \div, \sin, \exp$) to represent formula, offering more flexibility and requiring less prior knowledge than TP-SINDy's linear combination of functions in predefined function library. SymDL and NASSymDL use the internal functions (Cranmer et al., 2020) from the proxy models as supervision to compute fitness in genetic search. D-CODE uses the interpolated trajectories as supervision. TP-SINDy is based on sparse regression and uses estimated derivatives for symbolic regression, which can be noisy and inaccurate over large time intervals. CGS uses (3) from proxy model as references for search coordination and interpolated trajectories for fitness evaluation.

While our method co-evolves separate populations, it is fundamentally distinct from general-purpose techniques like the Cooperative Co-evolutionary Genetic Algorithm (CCGA) (Potter & De Jong, 1994). CCGA is a generic optimization framework that decomposes a problem and evolves sub-populations in a fixed, round-robin schedule, agnostic to the individual performance of each component. In stark contrast, CGS is specifically designed to solve the critical problem of symbolic overfitting in network dynamics. Its core innovation is an adaptive, reference-guided coordination strategy. Instead of following a blind, fixed schedule, CGS strategically prioritizes the evolution of the dynamic component (node or edge) that deviates most from a reliable neural reference. This targeted approach is crucial for discovering independently correct and physically meaningful governing equations—a challenge that CCGA's undirected, general-purpose search is not designed to address.

## 2.5 THEORETICAL ANALYSIS

Our theoretical analysis (see full proof in Appendix C) clarifies why CGS is more robust than a Separate Search (SS) baseline such as SymDL. We show that for SS to succeed, it requires a much stronger condition: the neural proxy must have small component-wise errors for both node and edge dynamics. In contrast, CGS only requires the overall trajectory error of the proxy to be small. This distinction aligns with the central motivation of our paper: CGS is resilient to the kind of overfitting where errors in the proxy's components compensate for each other, while SS is not. This is formalized in the following theorem.

**Theorem 1** (Error Bounds for CGS vs. SS). *Let $X_{gt}(t)$ be the ground-truth trajectory. Under Assumption 2 (see Appendix C), the ground-truth fitting errors for CGS ($E_{CGS}$) and Separate Search ($E_{SS}$) are bounded as follows:*

$$E_{CGS} = \|X_{F_{CGS}, G_{CGS}}(t) - X_{gt}(t)\| \leq \delta_{CGS} + \epsilon_{proxy} \tag{9}$$

$$E_{SS} = \|X_{F_{SS}, G_{SS}}(t) - X_{gt}(t)\| \leq L_F(\delta_F + \epsilon_F) + L_G(\delta_G + \epsilon_G) \tag{10}$$

*where $\epsilon_{proxy}$ is the neural proxy's overall trajectory error, while $\epsilon_F$ and $\epsilon_G$ are the errors of its individual node and edge dynamic components. The $\delta$ terms represent search algorithm errors, and $L_F, L_G$ are Lipschitz constants that reflect the sensitivity of the system to changes in $F$ and $G$.*

**Implication of the Theorem.** The theorem highlights a key difference: for SS to achieve a small error, it requires both $\epsilon_F$ and $\epsilon_G$ to be small—a much stronger condition than simply requiring a small $\epsilon_{proxy}$ as in CGS. In practice, overfitting often leads to large component-wise errors that could cancel out in the overall trajectory, so the SS bound can be much looser than the CGS bound. Thus, CGS is more robust to this type of overfitting, which is central to the motivation of our approach. See details of assumptions (and its Justification), theorem proof, and implications in Appendix C.

## 3 EXPERIMENTS

### 3.1 EXPERIMENTS ON SYNTHETIC DATASET

**Baseline**   We compare our method with baselines SymDL (Cranmer et al., 2020), SINDy (Brunton et al., 2016) and Two-Phase SINDy (TP-SINDy)(Gao & Yan, 2022). SymDL uses our proxy model and search the formulas of node and edge dynamics separately. SINDy (Brunton et al., 2016) is a sparse regression methods to find symbolic dynamics. SINDy here first numerically estimates the derivative of each node's activity through the five-point approximation (Sauer, 2011) and then optimizes the coefficients of the linear combination of predefined candidate functions. TP-SINDy is an improved version of it, which contains more elementary functions and an extra finetuning phase to remove terms with small coefficients. NASSymDL (Shi et al., 2023) is not included in the comparison because we leverage known inductive biases, negating the need for neural architecture search for the proxy model. We also do not compare against D-CODE (Qian et al., 2022) because it does not have a natural extension to the dynamics regression of multiple trajectories.

**Dataset**   We investigate the following four network dynamics in experiments, i.e., Susceptible-Infected-Susceptible (SIS) Epidemics Dynamics (Pastor-Satorras et al., 2015), Lotka-Volterra (LV) Population Dynamics (MacArthur, 1970), Wilson-Cowan Neural Firing Dynamic (Laurence et al., 2019; Wilson & Cowan, 1972) and Kuramoto Oscillators(KUR) model (Kuramoto & Kuramoto, 1984). Their dynamics are shown in Table 4. We conduct experiments on two complex network structures, i.e., Erdős-Rényi (ER) graph (Erdos & Renyi, 1959) and Barabási-Albert (BA) graph (Barabási & Albert, 1999) with 200 nodes.

We randomly initialize the state of all nodes and regularly sample $K$ timestamps $t_0, t_1, \cdots, t_{K-1}$ from the range $[0, T]$ because all other baselines rely on the equal time interval to estimated time derivatives. Then we simulate the whole dynamics to get the node states $[\mathbf{X}(t_0), \mathbf{X}(t_1), ..., \mathbf{X}(t_{K-1})]$. The edge weight $a_{vu}$ is set to binary values, i.e., $a_{vu} = 1$ if there is an edge between node $v$ and node $u$, otherwise $a_{vu} = 0$.

**Evaluation metrics**   The performance is evaluated by two metrics. (a) The recovery probability (**Rec. Prob.**) of formulas with correct skeletons. (See Appendix A for computation details). (b) The mean squared error (**MSE**) between the simulated trajectories using the recovered symbolic expression of dynamics and the ground truth observations. To ensure a fair evaluation, we only compute MSE for the symbolic expressions with correct skeletons.

**Results**   The comparison results are shown in Table 2. The proposed CGS generally has a higher recovery probability. For SIS and LV dynamics, TP-SINDy is not stable enough to recover the formula with the correct skeleton. This instability may stem from the numerical derivative estimation and a failure to effectively narrow down the model space. The latter can be exacerbated by data normalization, which may lead to overfitting candidate functions. For the WC dynamics, the TP-SINDy always fails to regress the correct skeleton, this is because the edge dynamics evolves a parametric function that cannot be represented by a linear combination of predefined functions. For the KUR dynamics, both TP-SINDy and CGS succeed with recovery probability 1. SINDy does not contain the finetuning phase which TP-SINDy has. As a result, it exhibits a lower recovery probability compared to TP-SINDy. SymDL's symbolic regression process relies solely on the proxy model while CGS utilizes two reference and interpolated trajectories for searching. As a result, CGS achieves the highest recovery probability. Refer to Appendix B.6 for additional results on multi-dimensional dynamics.

### 3.2 EXPERIMENTS ON REAL DATASET

**Dataset**   We demonstrate the effectiveness of CGS on the real epidemic network using the same influenza A (H1N1) spreading dataset as (Gao & Yan, 2022). In this dataset, each node represents a country or region, with the daily counts of newly reported cases as node states. The edges of the complex network are defined by the global aviation routes, depicting human mobility between regions. Our goal is to uncover the dynamics that govern the spread of the disease. For a fair comparison, we employed the same data preprocessing procedures as (Gao & Yan, 2022), such as constructing the adjacency matrix and data cleaning.

Table 2: Performance comparison on synthetic datasets. MSE values are scaled by $10^{-2}$ and multiply by $10^{-2}$ to obtain the actual values. (TP: TP-SINDy, PI: CGS, NA: MSE is not applicable because of failure of the correct skeleton recovery.)

| Graphs | Dynamics | Rec. Prob.↑ | | | | MSE↓ ($10^{-2}$) | | | |
|---|---|---|---|---|---|---|---|---|---|
| | | SymDL | SINDy | TP | PI | SymDL | SINDy | TP | PI |
| BA | SIS | 0.35 | 0.11 | 0.15 | 1.00 | 0.979±0.173 | 0.484±0.056 | 0.434±0.052 | 0.312±0.012 |
| | LV | 0.16 | 0.12 | 0.20 | 1.00 | 2.075±0.303 | 1.170±0.049 | 0.875±0.057 | 0.136±0.008 |
| | KUR | 0.80 | 0.87 | 1.00 | 1.00 | 0.064±0.018 | 0.175±0.016 | 0.040±0.003 | 0.007±0.001 |
| | WC | 0.56 | 0.00 | 0.00 | 1.00 | 0.362±0.057 | NA | NA | 0.092±0.004 |
| ER | SIS | 0.31 | 0.11 | 0.17 | 1.00 | 1.173±0.095 | 0.468±0.059 | 0.386±0.051 | 0.119±0.025 |
| | LV | 0.15 | 0.09 | 0.19 | 1.00 | 1.784±0.236 | 0.941±0.041 | 0.763±0.077 | 0.251±0.007 |
| | KUR | 0.87 | 0.78 | 1.00 | 1.00 | 0.071±0.018 | 0.087±0.022 | 0.069±0.019 | 0.017±0.001 |
| | WC | 0.40 | 0.00 | 0.00 | 1.00 | 0.266±0.047 | NA | NA | 0.044±0.003 |

**Results** We use CGS, SymDL, and TP-SINDy to regress the symbolic expression of influenza A spread dynamics. The result of CGS is

$$\dot{\mathbf{x}}_v(t) = a\mathbf{x}_v(t) + \sum_{u \in N_v} \frac{b}{1 + \exp - (m\mathbf{x}_v(t) + c)} \mathbf{x}_u(t), \tag{11}$$

where $a = 0.0740$, $b = 0.0015$, $m = -0.0041$ and $c = 9.9643$. Node dynamics in (11) is a linear function, aligning with the exponential growth of the epidemic. Edge dynamics in (11)

Figure 2: Visualizing the predicted number of newly reported cases in two regions using symbolic expressions from TP-SINDy and CGS.

is proportional to the neighboring region's state, which is consistent with the fact that the epidemic spreads increases with the number of infected cases in neighboring regions. The other factor of edge dynamics consists of a composition of a linear transformation followed by a sigmoid activation. This suggests that the rate of new infections from neighbors is modulated by the local infection level, possibly due to factors like population saturation or implemented control measures. This trend may be caused by the reduction of the willingness of people to travel to epidemic areas or the decrease of basic reproduction number ($R_0$) under high infected density.

The symbolic expression regressed by TP-SINDy and SymDL are

$$\dot{\mathbf{x}}_v(t) = a'\mathbf{x}_v(t) + \sum_{u \in N_v} \frac{b'}{1 + \exp - (\mathbf{x}_v(t) - \mathbf{x}_u(t))}, \tag{12}$$

$$\dot{\mathbf{x}}_v(t) = a''\mathbf{x}_v(t)^2 + \sum_{u \in N_v} \frac{b''}{1 + \exp - (\mathbf{x}_v(t) - \mathbf{x}_u(t))} \mathbf{x}_u(t) \tag{13}$$

where $a' = 0.074, b' = 7.130, a'' = 0.0742, b'' = 0.0012$. (12) fails to accurately model epidemic spreading, as it predicts a non-zero growth rate even without infected cases, which is physically unreasonable. In contrast, (11) correctly yields a zero growth rate in such scenarios, aligning with the fact that epidemics cannot spread without infected individuals. Compare with (13) from SymDL, (11) has a simpler node dynamics and edge dynamics, indicating that CGS can find more concise symbolic expressions.

We compare the trajectories of symbolic expressions in (11), (12) and (13) with the real infected cases. Fig. 2 visualizes the simulation results of inferred dynamics in two regions, i.e., "Finland" and "Saint Pierre and Miquelon". The trajectories of the infected cases in the two regions inferred by CGS are consistent with the ground truth, while the trajectories inferred by TP-SINDy and SymDL deviate from the ground truth.

We evaluate the errors of regressed dynamics. As the scale of infected cases varies across different regions, we normalize the infected cases to the range of $[0, 1]$ by the maximal value of each region. TP-SINDy's MSE (0.9028) exceeds CGS's (0.8261), demonstrating its superior fit to real data.

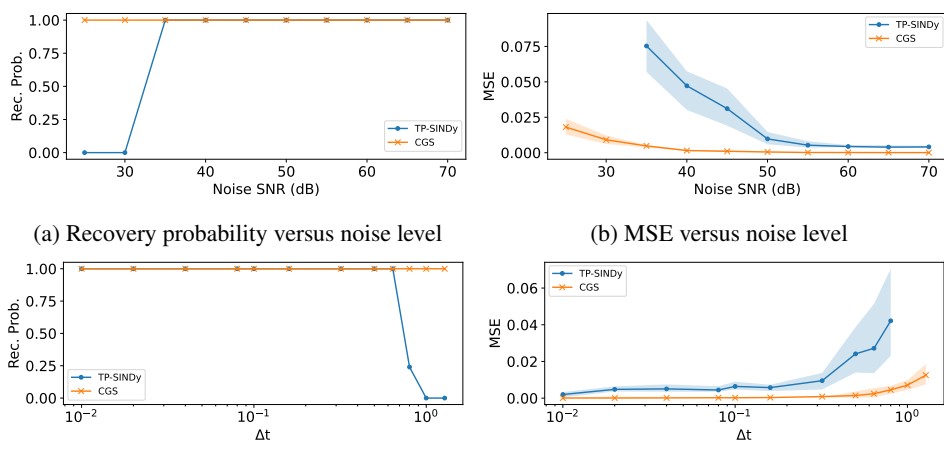

(a) Recovery probability versus noise level (b) MSE versus noise level

(c) Recovery probability versus time interval (d) MSE versus time interval

Figure 3: Evaluation of robustness. Shaded areas correspond to 95% confidence interval. (a) and (b) show the recovery probability and MSE when adding noise to observations. (c) and (d) show the recovery probability and MSE when increasing the time interval between observations.

### 3.3 ROBUSTNESS ON CGS AND TP-SINDY

We evaluate the robustness of CGS and TP-SINDy with the KUR dynamics, highlighting the advantages of the neural-symbolic approach over methods based on numerical derivatives (as stated in the introduction). We focus on performance under noisy observations and with large time intervals.

We add Gaussian noise to node states to assess performance under noise, with noise magnitude measured by the signal-to-noise ratio (SNR). As shown in Fig. 3(a), our method maintains a 100% recovery rate even as the SNR drops from 70 dB to 25 dB, while TP-SINDY's recovery rate falls to 0% at 30 dB. In Fig. 3(b), CGS consistently produces more accurate symbolic expressions that have lower MSE. This occurs because TP-SINDy relies on numerically estimating time derivatives that are noisy and inaccurate, whereas our method uses neural dynamics to denoise and interpolate observations directly. Deep neural networks excel at handling noisy data by learning meaningful patterns from large amounts of data, even when the data contains significant noise. Using the accurately denoised observations, CGS predicts constants in the formula better and produce a more accurate trajectory when noise exists.

TP-SINDy relies on the equal time interval to estimate time derivatives. So we increase the interval size to compare the performances of TP-SINDy and CGS. Sampling timestamps from $[0, 100]$ with different intervals $\Delta t$, Fig. 3(c) shows that our model maintains a $100\%$ recovery rate, while TP-SINDy fails with larger intervals. Fig. 3(d) shows that CGS always produces more accurate results when both methods produce the correct skeleton of dynamics. This is because the interpolated observations in CGS are better suited when the time interval is large compared to the estimated time derivatives used by TP-SINDy. Visualization of interpolated trajectories and estimated time derivatives are shown in Fig. 4 of the appendix.

## 4 RELATED WORK

### 4.1 SYMBOLIC REGRESSION

Throughout the history of physics, extracting elegant symbolic expressions from extensive experimental data has been a fundamental approach to uncovering new formulas and validating hypotheses. Symbolic Regression (SR) is a notable topic in this context (Schmidt & Lipson, 2009; Petersen et al., 2019; Cranmer et al., 2020; Kamienny et al., 2022), aiming to mimic the process of deriving an explicit symbolic model that accurately maps input $X$ to output $y$ while ensuring the model remains concise. Traditional methods for deriving formulas from data have predominantly relied on genetic programming (GP) (Schmidt & Lipson, 2009; Koza, 1994; Worm & Chiu, 2013), a technique inspired

by biological evolution that iteratively evolves populations of candidate solutions to discover the most effective mathematical representations.

More recently, due to the remarkable accomplishments of neural networks across diverse domains, there has been an increasing interest in leveraging deep learning for symbolic regression. Specifically, some recent works (Cranmer et al., 2020; Chen et al., 2021; Qian et al., 2022; Udrescu & Tegmark, 2020; Martius & Lampert, 2016; Mundhenk et al., 2021; Shi et al., 2023) have explored guiding genetic programming with the output of neural networks to improve the efficiency and accuracy of symbolic regression. This approach takes advantage of the powerful pattern recognition and generalization capabilities of neural networks to inform the evolutionary processes of genetic programming, resulting in more effective and efficient discovery of symbolic expressions. Another line of works (Kamienny et al., 2022; Biggio et al., 2021; Valipour et al., 2021; d'Ascoli et al., 2024) applies Transformer to symbolic regression and achieves comparable performance to GP-based methods.

### 4.2 Complex Network Dynamics Learning

Learning complex network dynamics from data has largely followed two paths: neural network-based approaches and symbolic regression.

Neural network-based methods often utilize an encode-process-decode paradigm (Hamrick et al., 2018; Zang & Wang, 2020), where initial node states are encoded, processed by a Graph Neural Network (GNN) to model evolution and interaction, and then decoded. For example, Murphy et al. (2021) used GNNs for regularly sampled observations, while NDCN (Zang & Wang, 2020) integrated graph neural ODEs (Chen et al., 2018) for continuous dynamics. While powerful, these models are typically black-boxes, limiting interpretability. Notably, Liu et al. (2023) recently achieved improved long-term prediction by dropping the encode-process-decode paradigm.

Symbolic regression aims for interpretable symbolic expressions. TP-SINDy (Gao & Yan, 2022), an extension of SINDy (Brunton et al., 2016), employs a broader function library and a two-phase regression. However, it relies on accurate time derivative estimates and is restricted to its predefined function library. Our work differentiates itself by using genetic programming, guided by supervision from neural dynamics, to discover symbolic expressions for complex network dynamics.

## 5 Conclusion

We introduced Coordinated Genetic Search (CGS), a novel algorithm that addresses the overfitting problem in symbolic regression of complex network dynamics, which often arises from searching for node and edge dynamics independently. CGS trains a neural proxy model to provide reliable references and then co-evolves two symbolic populations, coordinating their search by prioritizing the population that deviates more from its neural reference. This cooperative process steers the search toward a balanced, accurate solution. Evaluations on synthetic and real-world data show CGS significantly outperforms existing methods in formula recovery, prediction accuracy, and robustness, yielding simpler, more generalizable, and physically faithful models.

**Limitations** Our method has several limitations:

- CGS cannot successfully recover symbolic expressions when the formulations are highly complex or the dimension of node states is high. The highly complex formulations indicate a large search space for the genetic algorithm. Therefore, there should be a large population size and a large number of generations for the genetic algorithm to find the symbolic expressions. Since the fitness of our method is calculated based on the pairwise combination of node and edge dynamics, the fitness evaluation is computationally expensive and memory-consuming.

- CGS cannot deal with the complex network dynamics when some variables are missing in the observations. In some complex systems, it is difficult to observe all variables at the same time. In this case, the prediction accuracy of neural dynamics (4) may not be high enough to provide high-quality supervision data for symbolic regression.

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

Table 3: Comparison with different methods for symbolic regression. (GN: graph network, NAS: neural architecture search, GS: genetic search)

| Category | Design | SymDL | NASSymDL | D-CODE | TP-SINDy | CGS |
|---|---|---|---|---|---|---|
| Input | | input-output pairs | input-output pairs | single trajectory | multiple trajectories | multiple trajectories |
| Proxy model | Model design | GN w/ inductive bias | NAS | any regressor | – | GN w/ inductive bias |
| | Dynamics fitting data | estimated derivatives | estimated derivatives | raw observations | – | raw observations |
| Formula regression | Prior knowledge | elementary operation | elementary operation | elementary operation | function library | elementary operation |
| | Method | GS | GS | GS | sparse regression | coordinated GS |
| | Supervision | internal functions | internal functions | interpolated trajectory | estimated derivatives | network ref & interp. trajectories |
| Output | | input-output mapping | input-output mapping | ODE | Graph ODE | Graph ODE |

# A  DETAILS ON EXPERIMENTS

All experiments are implemented with PyTorch (Paszke et al., 2019), PyTorch Geometric (Fey & Lenssen, 2019), and gplearn (Stephens, 2015) in NVIDIA GeForce RTX 4090 GPUs and AMD EPYC 7763 Processors.

## A.1  DATASET STATISTICS

The BA graph is generated with 200 nodes and the initial degree of each node is set to 3. The ER graph is generated with 200 nodes and the probability for edge creation is set to 0.02. The initial states of SIS, LV, and WC dynamics are generated by randomly sampling from $[0, 1]$. For KUR dynamics, the initial states are generated by randomly sampling from $[0, 2\pi]$. For SIS dynamics, we set $\delta = 0.5$. For LV dynamics, we set $\alpha = 0.75$, $\theta = 0.5$,. For KUR dynamics, we set $\omega = 0.75$. For WC dynamics, we set $\tau = 0.75$, $\mu = 0.5$. We regularly sample 100 timestamps from $[0, 1]$ and simulate the dynamics to generate the observation data.

Table 4: Dynamics for generating synthetic dataset.

| | node dynamics | edge dynamics |
|---|---|---|
| SIS | $-\delta x_i(t)$ | $(1 - x_i(t))x_j(t)$ |
| LV | $x_i(t)(\alpha - \theta x_i(t))$ | $-x_i(t)x_j(t)$ |
| WC | $-x_i(t)$ | $(1 + \exp(-\tau(x_j(t) - \mu)))^{-1}$ |
| KUR | $\omega$ | $\sin(x_i(t) - x_j(t))$ |

## A.2  DETAILS FOR NETWORK TRAINING

We split the timestamps randomly into training, validation, and testing sets with a ratio of 0.8, 0.2, 0.1 to train the NeuralODE. We train the neural dynamics for 1000 epochs using optimizer AdamW. The learning rate is searched in the range of $[1e-3, 1e-2]$, the weight decay is set to $0.001$. We use MLPs as the encoder and decoder of neural dynamics. The hidden dimension of the neural dynamics is set to 10. The details of the network structure are shown in Table 5.

## A.3  DETAILS OF GENETIC SEARCH

We implement the coordinated genetic search based on gplearn (Stephens, 2015). gplearn (Stephens, 2015) represent the symbolic expressions as a syntax tree, where the functions are interior nodes, and the variables and constants make up the leaves. Evolution such as crossover, mutation, and reproduction are performed on the syntax tree. The population size of $\mathcal{F}$ and $\mathcal{G}$ are set to 200. The maximum generation of the genetic search $M$ is set to 50 and the stopping threshold $\epsilon = 10^{-5}$.

Table 5: Details of network structure for different dynamics.

|                        | SIS | LV | KUR | WC | real dataset |
|------------------------|-----|-----|-----|------|--------------|
| Hidden dimension       | 10  | 10  | 10  | 10   | 10           |
| Activation of $\phi^n$ | ReLU | ReLU | ReLU | ReLU | Sigmoid |
| Activation of $\phi^e$ | ReLU | Tanh | Tanh | Sigmoid | Sigmoid |
| Activation of Encoder  | ReLU | Tanh | Tanh | ReLU | Tanh |
| Activation of Decoder  | ReLU | Tanh | Tanh | ReLU | Tanh |
| Layer of $\phi^n$      | 2   | 2   | 1   | 1    | 2            |
| Layer of $\phi^e$      | 2   | 2   | 3   | 2    | 3            |

The $K$ in Algorithm 1 equals to 20. The function set includes addition, subtraction, multiplication, division, sine, cosine, and exponential. The constants are constrained in the range $[-1, 1]$. Other hyperparameters of gplearn are set as: p_crossover=0.6, p_subtree_mutation=0.1, p_hoist_mutation=0.05, p_point_mutation=0.1, parsimony_coefficient=0.01. We conduct the genetic search in 256 parallel threads to speed up the search process. Our CPUs are two AMD EPYC 7763 Processors.

### A.4 COMPUTATIONAL DETAILS OF REC. PROB.

The recovery probability is calculated as the ratio of the number of successful recovery of formula skeletons to the total number of experiments. We automatically check the correctness of the recovered formula skeletons using the method for verifying skeletons provided in Qian et al. (2022). Basically, we replace the constants in the formulas with placeholders and use the $\mathsf{simplify}(f' - f) == 0$ criterion from the Sympy package to determine if the skeleton is correct.

### A.5 DISCUSSIONS ON THE CHOICE OF METRICS

**Compute MSE between trajectories instead of the constants.**   We do not directly compute the MSE between predicted and true constants. This is because our goal is to evaluate how well the obtained symbolic expressions predict trajectories, which is crucial for real-world scenarios like epidemic forecasting. Directly computing constant errors is insufficient, as different constants impact the trajectory differently. Some constants require high precision, with small deviations causing significant errors, while others are less critical and can tolerate some errors.

**Compute MSE for formulas with correct skeletons.**   For simulated datasets, we choose MSE restricted to correctly recovered skeletons because the baseline methods often exhibit large MSE when recovering incorrect skeletons. Filtering out these formulas allows the baselines to achieve comparable performance. For real datasets, since the true dynamics skeleton is unknown, we directly compare the MSE of the trajectories without filtering by the skeletons.

## B ADDITIONAL RESULTS

### B.1 ABLATION STUDY

We conduct ablation studies to demonstrate the importance of interpolated trajectories in CGS. So, we test a variant of CGS which uses the original observations instead of interpolated and denoised trajectory when calculating the fitness. (CGS without Interp.).

Table 6 shows the results of SIS and LV dynamics in the BA graph. Without the interpolated and denoised observations, both the recovery probability and the accuracy of CGS drop. This indicates that the interpolated and denoised trajectories can provide high-quality fitness evaluation for symbolic regression.

We also experiment on the robustness of the ablation variants. Table 7 shows the results of KUR dynamics in the BA graph when the observations are noisy or the time interval is large. Different from the results in Table 6, the success Prob. significantly drop when removing the interpolation part. This proves the effectiveness of neural dynamics in denoising and augmenting trajectories.

Table 6: Ablation study with experiment results on SIS and LV dynamics in BA graph.

| Model | SIS | | LV | |
|---|---|---|---|---|
| | Rec. Prob.↑ | MSE↓ $(10^{-2})$ | Rec. Prob.↑ | MSE↓ $(10^{-2})$ |
| CGS | 1 | 0.312±0.012 | 1 | 0.136±0.008 |
| CGS w/o Interp. | 0.81 | 0.408±0.027 | 0.86 | 0.588±0.028 |

Table 7: The robustness of two variants compared with full method on KUR dynamics in BA graph.

| Models | Noise (SNR=35dB) | | Time interval $(\Delta t = 1.28)$ | |
|---|---|---|---|---|
| | Rec. Pro.↑ | MSE↓ $(10^{-2})$ | Rec. Pro.↑ | MSE↓ $(10^{-2})$ |
| CGS | 1 | 0.478±0.103 | 1 | 0.454±0.213 |
| CGS w/o Interp | 0.84 | 6.970±1.870 | 0.78 | 2.645±1.138 |

## B.2 RUNTIME

In Table 8, CGS saves 30.1% running time on SIS dynamics and 39.0% running time on LV dynamics compared with CGS w/o Coord. (SymDL). The results show that the coordinated genetic search can significantly reduce the search space and improve the efficiency of the search process.

Table 8: The runtime (minutes) of CGS and CGS(w/o Coord.).

| Model | SIS | LV |
|---|---|---|
| CGS | 61.5 | 50.9 |
| CGS w/o Coord. (SymDL) | 88.0 | 83.4 |

## B.3 EXAMPLES OF EXPRESSIONS FROM SYMBOLIC EXPRESSIONS

In this section, we provide examples of symbolic expressions of CGS, TP-SINDy (Rec.), TP-SINDy (Fail), SymDL (Rec.) and SymDL (Fail) on SIS, LV, KUR, and WC dynamics in the BA graph. TP-SINDy (Rec.) represents the symbolic expressions of TP-SINDy when the skeleton of the dynamics is successfully recovered, while TP-SINDy (Fail) represents the symbolic expressions of TP-SINDy when the skeleton of the dynamics is not successfully recovered. SymDL (Rec.) and SymDL (Fail) are the results of SymDL with correct/incorrect skeletons. The expressions are shown in Table 9.

## B.4 EXAMPLE OF OVERFITTING

Take the SIS dynamics in the BA graph as an example. The symbolic expressions of CGS, TP-SINDy (Rec.), TP-SINDy (Fail), SymDL (Rec.) and SymDL (Fail) are shown in Table 9. We compute the MSE of the predicted trajectories under interpolated and extrapolated settings. The results are shown in Table 10. Although the symbolic expressions from baseline methods have relatively low MSE values under the interpolated setting, their MSE values are much higher under the extrapolated setting. This indicates that the symbolic expressions are overfitted and cannot generalize well to extrapolated setting. We double the time range from $[0, 1]$ to $[0, 2]$ to evaluate the extrapolation performance.

Note that all failure cases in Table 9 can also be viewed as examples of overfitting. The symbolic expressions of CGS are more interpretable and simpler while the overfitted symbolic expressions of TP-SINDy/SymDL are more complex and contain more terms.

## B.5 VISUALIZATION OF NEURAL DYNAMICS

When observations are noisy or time interval is large, the neural dynamics can denoise and interpolate the observations to provide high-quality supervision data for symbolic regression. On the other hand, the numerical estimation is sensitive to noise and needs the sample interval to be small enough. We

Table 9: Symbolic regressions of CGS, TP-SINDy (Rec.), TP-SINDy (Fail) , SymDL (Rec.) and SymDL (Fail) on SIS, LV, KUR, and WC dynamics in the BA graph.

| Dynamics | Models | Node dynamics | Edge dynamics |
|---|---|---|---|
| | GT | $-0.5x_i(t)$ | $(1 - x_i(t))x_j(t)$ |
| SIS | CGS | $-0.48540x_i(t)$ | $(1 - x_i(t))x_j(t)$ |
| | TP-SINDy (Rec.) | $-0.46640x_i(t)$ | $(0.99119 - 1.09637x_i(t))x_j(t)$ |
| | TP-SINDy (Fail) | $-0.47256x_i^2(t) - 0.12596$ | $0.10860\text{sigmoid}(x_j(t) - x_i(t)) +$ $0.18835(x_j(t) - x_i^2(t))$ $0.19917(x_j(t) - x_i(t)) + 0.35416\sin(x_j(t))$ |
| | SymDL (Rec.) | $-0.54827x_i(t)$ | $(1.03945 - 0.92538x_i(t))x_j(t)$ |
| | SymDL (Fail) | $-0.6017x_i(t)\sin(\cos(x_i(t) - 0.5178))$ | $(0.9966 - 1.1798x_i(t) + 0.1984\sin(x_i(t)))x_j(t)$ |
| | GT | $x_i(t)(0.75 - 0.5x_i(t))$ | $-x_i(t)x_j(t)$ |
| LV | CGS | $x_i(t)(0.75034 - 0.48812x_i(t))$ | $-0.99428x_i(t)x_j(t)$ |
| | TP-SINDy (Rec.) | $x_i(t)(0.69882 - 0.41853x_i(t))$ | $-0.91701x_i(t)x_j(t)$ |
| | TP-SINDy (Fail) | $0.03984 + 0.36330 * \sin(x_i(t))$ | $-0.945810x_i(t)x_j(t) - 0.11895x_i(t)x_j^2(t)$ |
| | SymDL (Rec.) | $x_i(t)(0.703971 - 0.54885x_i(t))$ | $-1.11962x_i(t)x_j(t)$ |
| | SymDL (Fail) | $(x_i(t) - 0.027459) * \exp(-1.2791x_i(t))$ | $-1.006(x_i(t) - 0.0031034)x_j(t)$ |
| | GT | $0.75$ | $\sin(x_i(t) - x_j(t))$ |
| KUR | CGS | $0.75002$ | $\sin(1.0001x_i(t) - x_j(t))$ |
| | TP-SINDy (Rec.) | $0.75014$ | $0.99899\sin(x_i(t) - x_j(t))$ |
| | TP-SINDy (Fail) | NA | NA |
| | SymDL (Rec.) | $0.74777$ | $0.99037\sin(x_i(t) - x_j(t))$ |
| | SymDL (Fail) | $x_i(t) * 0.00815 + 0.74624$ | $0.99725\sin(x_i(t) - 1.003x_j(t) + 0.0013507)$ |
| | GT | $-x_i(t)$ | $\text{sigmoid}(-0.75(x_j(t) - 0.5))$ |
| WC | CGS | $-x_i(t)$ | $\text{sigmoid}(-0.74503(x_j(t) - 0.49128))$ |
| | TP-SINDy (Rec.) | NA | NA |
| | TP-SINDy (Fail) | $-0.82267x_i(t)$ | $0.08513\text{sigmoid}(x_j(t) - x_i(t))$ $+0.68484\text{sigmoid}(x_j(t))$ |
| | SymDL (Rec.) | $-1.0132x_i(t)$ | $\text{sigmoid}(-0.78165(x_j(t) - 0.47162))$ |
| | SymDL (Fail) | $-1.0953x_i(t) - 0.00437$ | $\text{sigmoid}(\text{sigmoid}((x_i(t) + 0.7432) * x_i(t)) - 0.046806)$ |

Table 10: The MSE of CGS, TP-SINDy (Rec.), TP-SINDy (Fail.) , SymDL (Rec.) and SymDL (Fail) under interpolated and extrapolated settings on SIS dynamics in the BA graph.

| | CGS | TP-SINDy (Rec.) | TP-SINDy (Fail.) | SymDL (Rec.) | SymDL (Fail.) |
|---|---|---|---|---|---|
| Interpolation | $3.2 \times 10^{-3}$ | $5.2 \times 10^{-3}$ | $147.2 \times 10^{-3}$ | $6.3 \times 10^{-3}$ | $244.6 \times 10^{-3}$ |
| Extrapolation | $3.9 \times 10^{-3}$ | $46.9 \times 10^{-3}$ | $697.0 \times 10^{-3}$ | $122.4 \times 10^{-3}$ | $1678.6 \times 10^{-3}$ |

visualize the interpolate trajectories and the estimated time derivatives in Fig. 4, which is consistent with our contributions.

## B.6 RESULTS FOR MULTI-DIMENSIONAL DYNAMICS

The proposed method can be applied to multi-dimensional dynamics. We test the performance of CGS on the FitzHugh–Nagumo (FHN) dynamics which are proposed to model the activity of neural systems (Rabinovich et al., 2006). The formula is shown in Table 11. The dimension 1 represents the membrane voltage and dimension 2 represents the recovery variable.

Table 11: Dynamics for FitzHugh-Nagumo dynamics.

| | node dynamics | edge dynamics |
|---|---|---|
| dimension 1 | $x_{i,1}(t) - x_{i,2}(t) - \frac{1}{3}x_{i,1}(t)^3$ | $x_{j,1}(t) - x_{i,1}(t)$ |
| dimension 2 | $ax_{i,1}(t) + bx_{i,2}(t) + c$ | $0$ |

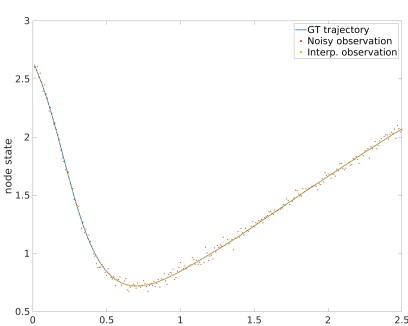
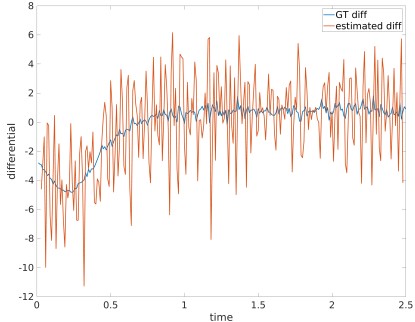

(a) Interpolated trajectory with noisy observation. (b) Estimated time derivative with noisy observation.

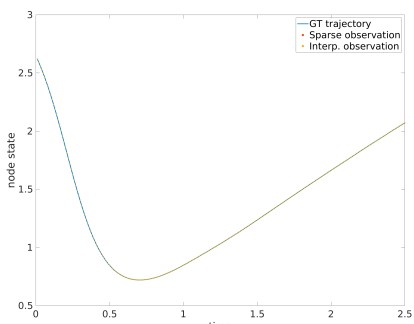
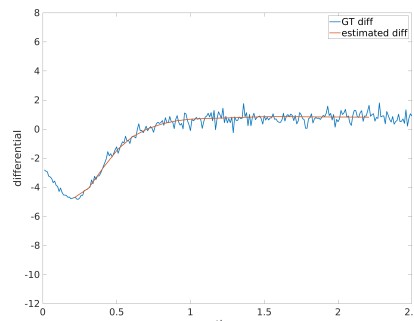

(c) Interpolated observations with large sample time interval.

(d) Estimated time derivative with large sample time interval.

Figure 4: Visualization of interpolated and denoised observations and the estimated time derivative. (a) The interpolated observations are very close to the ground truth when noise exists. (b) The estimated time derivative is inaccurate with noisy observation. (c) The interpolated observations are close to the ground truth with a large time interval (0.1). (d) The estimated time derivative is inaccurate when the sample time interval is large.

The neural network can be directly applied to multi-dimensional dynamics by extending its input dimensions. For the genetic search component, we adapt the existing package to support vector-valued functions. Gplearn (Stephens, 2015) represents scalar-valued functions using a syntax tree. In our approach, vector-valued functions are represented as a "syntax forest," which is a collection of syntax trees. Mutation and crossover operations are conducted independently for each dimension. This coordinated genetic search framework seamlessly extends to handle multi-dimensional dynamics.

We evaluate the performance of CGS on the FHN dynamics within a BA graph. CGS successfully reconstructs the dynamics' skeleton with a success probability of 1. Furthermore, it achieves a mean squared error (MSE) of $0.182 \times 10^{-2}$, outperforming TP-SINDy, which yields a higher MSE of $0.454 \times 10^{-2}$.

## C  PROOF OF THEORY

**Assumption 2.**    1. **Bounded Neural Proxy Error:** $\left\| \hat{X}(t) - X_{gt}(t) \right\| \leq \epsilon_{proxy}$, $d(\hat{F}, F_{gt}) \leq$

$\epsilon_F$, and $d(\hat{G}, G_{gt}) \leq \epsilon_G$ for small positive constants $\epsilon_{proxy}, \epsilon_F, \epsilon_G$.

2. **Lipschitz Continuity of Dynamics:** There exist constants $L_F, L_G > 0$ such that for any two pairs of dynamics $(F_1, G_1)$ and $(F_2, G_2)$,

$$\|X_{F_1,G_1}(t) - X_{F_2,G_2}(t)\| \leq L_F d(F_1, F_2) + L_G d(G_1, G_2)$$

Here, $d(f_1, f_2)$ denotes the average absolute error $|f_1 - f_2|$ between two functions.

3. **Bounded Search Error:** $\left\|X_{F_{CGS},G_{CGS}}(t) - \hat{X}(t)\right\| \leq \delta_{CGS}$, $d(F_{SS}, \hat{F}) \leq \delta_F$, and $d(G_{SS}, \hat{G}) \leq \delta_G$ for small positive constants $\delta_{CGS}, \delta_F, \delta_G$.

**Justification of Assumption 2**

- **Bounded Neural Proxy Error:** This assumption is standard in neural symbolic regression. It states that the neural proxy model provides a reasonably accurate approximation of the true dynamics, both in terms of the overall trajectory and the individual components. Importantly, our main theorem shows that CGS only requires the overall trajectory error $\epsilon_{proxy}$ to be small, while SS requires the much stronger condition that both $\epsilon_F$ and $\epsilon_G$ are small.

- **Lipschitz Continuity of Dynamics:** This assumption is common in the analysis of dynamical systems. The Lipschitz constants $L_F$ and $L_G$ quantify how sensitive the system's trajectory is to changes in the node and edge dynamics, respectively. If these constants are large, the error bounds become looser, meaning that small errors in $F$ or $G$ can lead to larger deviations in the trajectory.

- **Bounded Search Error:** This assumption is justified by the convergence properties of genetic algorithms (Rudolph, 1994), which shows that variations of the genetic algorithms that ensure the best solution in the population is always preserved are guaranteed to converge to the global optimum. As we ensure the best solution in the population is always preserved in CGS, such an assumption is reasonable.

**Theorem 3.** *Let the ground-truth dynamics of a complex network be described by $\dot{\mathbf{x}}_v(t) = F_{gt}(\mathbf{x}_v(t)) + \sum_{u \in \mathcal{N}_v} G_{gt}(\mathbf{x}_v(t), \mathbf{x}_u(t))$, which produces a ground-truth trajectory $X_{gt}(t)$. Let a neural proxy model $f_\theta$ be trained on observed data, producing neural references $\hat{F}$ and $\hat{G}$, and a denoised, interpolated trajectory $\hat{X}(t)$.*

*We define two search strategies to find symbolic expressions $(F, G)$:*

1. ***Separate Search (SS):** A baseline approach like SymDL that finds expressions $(F_{SS}, G_{SS})$ by independently minimizing the distance to the neural references: $F_{SS} = \arg\min_{F \in \mathcal{F}} d(F, \hat{F})$ and $G_{SS} = \arg\min_{G \in \mathcal{G}} d(G, \hat{G})$.*

2. ***Coordinated Genetic Search (CGS):** Finds expressions $(F_{CGS}, G_{CGS})$ by minimizing the trajectory error against the denoised trajectory: $(F_{CGS}, G_{CGS}) = \arg\min_{(F,G) \in \mathcal{F} \times \mathcal{G}} \left\|X_{F,G}(t) - \hat{X}(t)\right\|$, where $X_{F,G}(t)$ is the trajectory simulated using $(F, G)$.*

*Under Assumption 2, the ground-truth fitting error for CGS, $E_{CGS} = \|X_{F_{CGS},G_{CGS}}(t) - X_{gt}(t)\|$, is bounded by $E_{CGS} \leq \delta_{CGS} + \epsilon_{proxy}$. The error for SS, $E_{SS} = \|X_{F_{SS},G_{SS}}(t) - X_{gt}(t)\|$, is bounded by $E_{SS} \leq L_F(\delta_F + \epsilon_F) + L_G(\delta_G + \epsilon_G)$.*

*Proof.* We aim to establish upper bounds for the true fitting error $E = \|X_{F,G}(t) - X_{gt}(t)\|$ for both CGS and SS.

**Part 1: Bounding the Error of CGS** ($E_{CGS}$)   The fitting error for the solution found by CGS is $E_{CGS} = \|X_{F_{CGS},G_{CGS}}(t) - X_{gt}(t)\|$. Using the triangle inequality, we can introduce the denoised trajectory $\hat{X}(t)$ from the neural proxy:

$$E_{CGS} \leq \left\|X_{F_{CGS},G_{CGS}}(t) - \hat{X}(t)\right\| + \left\|\hat{X}(t) - X_{gt}(t)\right\| \tag{14}$$

By Assumption 2.3, the first term is the error minimized by the CGS algorithm, which is bounded by $\delta_{CGS}$. By Assumption 2.1, the second term is the trajectory error of the neural proxy model, which is bounded by $\epsilon_{proxy}$. Substituting these bounds into the inequality, we get:

$$E_{CGS} \leq \delta_{CGS} + \epsilon_{proxy} \tag{15}$$

This bound depends on the quality of the CGS search ($\delta_{CGS}$) and the overall accuracy of the neural proxy's integrated trajectory ($\epsilon_{proxy}$).

**Part 2: Bounding the Error of SS** ($E_{SS}$) The fitting error for the solution found by SS is $E_{SS} = \|X_{F_{SS},G_{SS}}(t) - X_{gt}(t)\|$. Since the ground-truth trajectory is generated by the true dynamics $(F_{gt}, G_{gt})$, we can write $X_{gt}(t) = X_{F_{gt},G_{gt}}(t)$. Thus, the error is:

$$E_{SS} = \left\|X_{F_{SS},G_{SS}}(t) - X_{F_{gt},G_{gt}}(t)\right\| \tag{16}$$

Using the Lipschitz continuity from Assumption 2.2, we can bound this trajectory error by the distance between the symbolic component functions:

$$E_{SS} \leq L_F d(F_{SS}, F_{gt}) + L_G d(G_{SS}, G_{gt}) \tag{17}$$

Now, for each component, we use the triangle inequality to introduce the neural references $\hat{F}$ and $\hat{G}$:

$$d(F_{SS}, F_{gt}) \leq d(F_{SS}, \hat{F}) + d(\hat{F}, F_{gt}) \tag{18}$$

$$d(G_{SS}, G_{gt}) \leq d(G_{SS}, \hat{G}) + d(\hat{G}, G_{gt}) \tag{19}$$

Substituting these back into the inequality for $E_{SS}$:

$$E_{SS} \leq L_F(d(F_{SS}, \hat{F}) + d(\hat{F}, F_{gt})) + L_G(d(G_{SS}, \hat{G}) + d(\hat{G}, G_{gt})) \tag{20}$$

By Assumption 2.3, the terms $d(F_{SS}, \hat{F})$ and $d(G_{SS}, \hat{G})$ represent the errors of the separate search process, bounded by $\delta_F$ and $\delta_G$ respectively. By Assumption 2.1, the terms $d(\hat{F}, F_{gt})$ and $d(\hat{G}, G_{gt})$ represent the component-wise errors of the neural proxy, bounded by $\epsilon_F$ and $\epsilon_G$. This gives the final bound:

$$E_{SS} \leq L_F(\delta_F + \epsilon_F) + L_G(\delta_G + \epsilon_G) \tag{21}$$

This bound depends on the quality of the separate search for each component ($\delta_F, \delta_G$) and accumulates the individual component errors from the neural proxy ($\epsilon_F, \epsilon_G$). □

**Implication of the Theory** The derived bounds show a fundamental difference. The CGS error is limited by the neural proxy's ability to denoise and predict the *overall trajectory*. The SS error is limited by the proxy's ability to accurately disentangle and identify the *individual dynamic components*.

As argued in the paper, a critical form of overfitting occurs when errors in the neural components, $\epsilon_F$ and $\epsilon_G$, are large but compensate for one another during integration, resulting in a small trajectory error $\epsilon_{proxy}$. In such a scenario, $L_F\epsilon_F + L_G\epsilon_G \gg \epsilon_{proxy}$. Consequently, the upper bound on the error for SS ($E_{SS}$) becomes significantly larger than that for CGS ($E_{CGS}$), formally demonstrating that the cooperative search strategy has a better fitting performance by avoiding the accumulation of component-wise errors.

## THE USE OF LARGE LANGUAGE MODELS (LLMS)

Large Language Models (LLMs) were used to assist with manuscript preparation and code implementation. All LLM-generated content was reviewed and edited by the authors to ensure accuracy.

