# OpenReview forum: "Coordinated Search for Symbolic Formulas of Complex Network Dynamics"
_ICLR.cc/2026/Conference — Submitted to ICLR 2026_

### Official Review · Reviewer_p1Da · 2025-10-30

**Soundness:** 2
**Presentation:** 3
**Contribution:** 2
**Rating:** 2
**Confidence:** 4

**Summary:**

This paper proposes an approach to improve symbolic regression for network systems. It focuses on the problem of ensuring separation in the discovered equations between the term modelling node self-dependence and neighbourhood dependence. For that matter, the authors propose a modification to the genetic search to favour the population with highest distance from neural proxies. The approach is evaluated on several synthetic network systems as well as on a real-world dataset of epidemic evolution.

**Strengths:**

- Clearly written paper
- Robustness and ablation analyses

**Weaknesses:**

- Lack of design choice motivations
- Lack of numerical evaluation details (e.g. choice of dictionaries for competitor methods, hyperparasites, number of runs of probability of success)
- Employs genetic programming which is not scalable
- The theoretical result is not convincing. To make the point that CGS is better than SS, one would need to show a lower bound on the error of SS that is higher or equal than the error bound of CGS
- Overhead of training neural proxies

**Questions:**

1/ Given that the neural proxies can suffer from compensation between the ODE components, how does neural proxy-based coordination lead to improved fitting of the symbolic expressions ?

2/ Which dictionaries of elementary functions were used for TP-SINDy ? Was the regularisation parameter tuned for the reported experimental results ?

3/ Could you elaborate on the importance of well separated ODE terms for real-world data, given that "all models are wrong, but many are useful" ?

---

### Official Review · Reviewer_drTB · 2025-10-31

**Soundness:** 3
**Presentation:** 3
**Contribution:** 2
**Rating:** 6
**Confidence:** 5

**Summary:**

Coordinated Genetic Search (CGS) offers a collaborative search approach for symbolic regression on complex networks, effectively addressing the overfitting problem that arises when optimizing node and edge dynamics independently. It demonstrates superior performance on both synthetic and real-world datasets, and holds high application value in fields such as network science and epidemiology.

**Strengths:**

1. The authors propose a novel search algorithm, Coordinated Genetic Search (CGS), designed to jointly evolve symbolic expressions for both node and edge dynamics.
2. Through theoretical analysis, the paper further demonstrates that CGS exhibits robustness to surrogate model errors.

**Weaknesses:**

1. The CGS algorithm requires fitness evaluation over combinations of symbolic expressions for node and edge dynamics (Eqs. 6–7). When the population size is large or the node dimensionality is high, the computational cost can become substantial.
2. Theorem 1 is derived under Assumption 3, which presumes bounded search error—i.e., that the genetic algorithm can converge to a satisfactory solution. However, genetic programming in practice does not guarantee global optimality.
3. The baselines used in the experiments—SymDL (2020), SINDy (2016), and TP (2022)—are relatively dated and do not reflect recent advances in complex system dynamics discovery. As a result, the current comparisons may not adequately demonstrate the superiority of the proposed method in modeling and uncovering complex dynamical laws.

**Questions:**

Q1. In Theorem 1, the error bound depends on the Lipschitz constants L_F and L_G. Do these constants vary significantly across different types of dynamical systems (e.g., SIS vs. Kuramoto)? How are such constants estimated or determined in practical applications?
Q2. Assumption 2 requires that the component errors of the surrogate models (ϵF, ϵG) are bounded. However, the main text mentions that error compensation may occur during surrogate training. Does this phenomenon conflict with the bounded-error assumption, and if so, do the theoretical results still hold under such conditions?
Q3. The coordination mechanism (Eq. 5) is presented as a core innovation of CGS, but no comparison is provided with alternative search strategies. Have the authors examined how such alternatives affect performance or stability, and what motivates the adoption of the current design?
Q4. The related work section provides a limited discussion of recent studies from the past two years. Could the authors expand this part to include recent developments in complex network dynamics, thereby clarifying the novelty of our approach?

---

### Official Review · Reviewer_Ajnd · 2025-10-31

**Soundness:** 3
**Presentation:** 2
**Contribution:** 2
**Rating:** 2
**Confidence:** 5

**Summary:**

The paper proposes a Coordinated Genetic Search (CGS) algorithm for discovering symbolic formulas of ODEs governing graph dynamical systems. The method combines a neural proxy model and a genetic search algorithm to recover interpretable governing equations. Specifically, the algorithm first trains a neural model on raw observations to derive denoised and interpolated trajectories. Then, this neural reference is used to guide the genetic search of the symbolic formulas for both self-interaction and edge dynamics, prioritizing the evolution of the population that deviates most from its neural reference. By co-evolving node and edge dynamics jointly, CGS addresses the problem of overfitting that may arises when searching for the two symbolic components independently. Experiments are conducted on synthetic and real-world datasets to demonstrate the efficacy of the proposed method compared to existing approaches.

Although the problem of learning symbolic representations of dynamical systems is relevant, the author's contribution is not supported by their assumptions and limited experimental evaluations. In particular, it lacks reproducibility and source code.

**Strengths:**

- The idea of coordinating the symbolic regression of the two components of dynamical systems is worth investigating.
- The paper follows traditional notation and reports many relevant works in this domain.
- Section 2.4 is nicely written.
- The appendix supports the main work with relevant information and analyses.

**Weaknesses:**

- The statement on how existing methods search for these components independently and "results in models that are neither interpretable nor generalizable, as illustrated in Table 1.", is not justified, and Table 1 does not really prove anything. For instance, basic symbolic regression approaches, SymDL, TPSINDy, are interpretable and generalizable.
- The main argument that training the components independently leads to overfitting is not supported by any proof or references. How exactly would the error in one component being compensated by the other be an issue? Couldn't the symbolic terms be merged in the symbolic expression? Or restrict G to learn interactions/edge dynamics, and F the rest?
- Section 2.2 is not clear. How is the symbolic population defined at this point? With "evolution" are you then referring to genetic programming? Do you compute the distances at each step?
- Section 3.1 Baseline is redundant with the previous subsection and section 4.
- Any reference to reproducibility and missing code makes this paper less trustworthy.

Minor comments that did not impact the score:
- There are many arXiv preprint references, which should be avoided. Refer to published versions if available, or other peer-reviewed contributions.
- The notations of the "°" and hat symbols are not explained.

**Questions:**

- In 2.1: "After the training, we will use the estimated node dynamics and edge dynamics as references for the coordinated search and use the interpolated trajectory [...] as the signal for fitness evaluation." Does it mean that both quantities are the results of learning and estimations, hence are not based on input data? This should be explained better, along with section 2.3.
- In (6) and (7), what is the error function? Is it the distance or a loss?
- Why did you split the timesteps randomly and not sequentially? Wouldn't this be a case of data leakage from future dynamical states?
- Why did you choose to report "Finland" and "Saint Pierre and Miquelon” in Fig. 2? Why not other countries?

---

### Official Review · Reviewer_xuFF · 2025-11-01

**Soundness:** 2
**Presentation:** 3
**Contribution:** 2
**Rating:** 4
**Confidence:** 4

**Summary:**

The paper proposes Coordinated Genetic Search (CGS), a neural-guided, co-evolution strategy that discovers both node and edge symbolic dynamics of networked systems together, to avoid the overfitting that occurs when they’re searched separately. Benchmarking on several problems and different noise settings proves the effectiveness of the proposed method in accuracy and robustness.

**Strengths:**

1. **Clear Problem Framing and Motivation**: Identifies the “compensation” pathology when learning node and edge dynamics separately, and proposes a reference-guided coordination rule that evolves only the population deviating more from its neural reference.

2. **Theoretical/Limitation Analyses**: Provides bounds showing CGS depends on the proxy’s overall trajectory error $\epsilon_{proxy}$ while Separate Search (SS) accumulates component-wise errors $\epsilon_F$ and $\epsilon_G$, supporting robustness to compensation. Limitations show the border cases where CGS might fail.

3. **Real-world Case with Interpretable Form**: On H1N1, discovers a linear node term and sigmoid-modulated edge term with qualitatively better fits in multiple regions, illustrating interpretability and plausibility.

**Weaknesses:**

1. **Potentially Prone to Proxy Quality and Assumptions.**:  Both coordination and theory require a reasonably accurate proxy (bounded $\epsilon_{proxy}$) and Lipschitz dynamics; if the proxy mis-models the system or error is uneven across state space, coordination choices and fitness may be biased.

2. **Potential Circularity in Supervision**: Fitness is computed against proxy-generated trajectories, not raw observations. If the proxy overfits or hallucinates structure, CGS could chase its artifacts despite theoretical safeguards.

3. **Experimental Design**: MSE is reported only when the ground truth expressions are fully recovered by PI in Table 2. The lack of MSE comparisons in complicated cases where all benchmarking methods cannot fully recover the ground truth solutions reduces comparative depth.

4. **Lack of Hyperparameter Sensitivity Analysis**: The performances may be sensitive to Big-K selection, parsimony penalty, limited function set, and constant range [−1,1]. No sensitivity study is provided.

**Questions:**

1. Since fitness is computed against proxy-generated, denoised/interpolated trajectories rather than raw observations, how do you guard against proxy bias? Do results change if you blend raw and interpolated signals or weaken the proxy?

2. What is the complexity of evaluating all cross-population pairs and selecting Big-K? Any heuristics to prune pairs early, and how sensitive are results to the choice K=20?

3. With population=200, 50 generations, and 256 threads, what’s the wall-clock per task, and how does cost scale with larger libraries/graphs? Any runtime comparison to a non-coordinated GP?

4. The proxy is trained on the same trajectories later used as fitness targets. Do you hold out time slices or nodes when evaluating GP fitness to avoid circularity?

---

### Meta-Review · Area_Chair_vvdq · 2026-01-03

**Summary:**

This submission presents a Coordinated Genetic Search (CGS) algorithm for symbolic regression discovering ODEs governing network dynamics. By neural-guided co-evolving node and edge dynamics with genetic search, CGS is claimed to avoid overfitting that may arises when searching for the two symbolic components independently. Experiments on synthetic and real-world datasets were presented to demonstrate the efficacy of CGS.

**Reviewer Concerns:**

Besides the suggestions on improving overall presentation and having more comprehensive/reproducible experiments, reviewers have the following concerns:

1. The claim of overfitting in the existing methods and the motivation that CGS can avoid overfitting may be better justified (Ajnd, p1Da).

2. The reviewers also raised concerns on scalability of CGS considering both the overheads of neural proxies and genetic programming (Ajnd, drTB, p1Da, xuFF).

3. The provided theoretical analyses may not provide clear understanding on the benefits of CGS (drTB, p1Da, xuFF).

4. Reviewer xuFF also raised the questions on the sensitivity of CGS results, particularly in the context of neural proxy performance variability.

5. Reviewer drTB requested benchmarking to more up-to-date baselines.

The authors can consider addressing these raised concerns to further improve their presented work.

**Reviewer Scores:**

The authors did not participate in the rebuttal discussion.

---

### Decision · Program_Chairs · 2026-01-26

Reject